# Sub-surface thermal measurement in additive manufacturing via machine learning-enabled high-resolution fiber optic sensing

Rongxuan Wang ®[1], Ruixuan Wang ®[2], Chaoran Dou[3], Shuo Yang[4], Raghav Gnanasambandam ®[5], Anbo Wang[2] & Zhenyu (James) Kong ®[3] ✉

Microstructures of additively manufactured metal parts are crucial since they determine the mechanical properties. The evolution of the microstructures during layer-wise printing is complex due to continuous re-melting and reheating effects. The current approach to studying this phenomenon relies on time-consuming numerical models such as finite element analysis due to the lack of effective sub-surface temperature measurement techniques. Attributed to the miniature footprint, chirped-fiber Bragg grating, a unique type of fiber optical sensor, has great potential to achieve this goal. However, using the traditional demodulation methods, its spatial resolution is limited to the millimeter level. In addition, embedding it during laser additive manufacturing is challenging since the sensor is fragile. This paper implements a machine learning-assisted approach to demodulate the optical signal to thermal distribution and significantly improve spatial resolution to 28.8 μm from the original millimeter level. A sensor embedding technique is also developed to minimize damage to the sensor and part while ensuring close contact. The case study demonstrates the excellent performance of the proposed sensor in measuring sharp thermal gradients and fast cooling rates during the laser powder bed fusion. The developed sensor has a promising potential to study the fundamental physics of metal additive manufacturing processes.

Laser powder bed fusion (L-PBF) is one of the most promising additive manufacturing techniques for fabricating metal parts with complex geometry and hard-to-process materials. However, achieving high-quality metal parts without defects using L-PBF is challenging. A non-homogenous microstructure is one of the crucial defects. Microstructures are the crystal structure of materials at the micro level, such as grains, misalignments, and grain boundaries. It is the intrinsically driven force of mechanical performance, such as the strength and hardness of materials. Therefore, it is crucial to understand the formation and evolution of microstructures of the metals during the L-PBF. Most existing work on microstructure formation focuses on a single track or point melting due to limited measurement and

[1]Department of Industrial and Systems Engineering, Auburn University, Auburn, AL, USA. [2]Bradley Department of Electrical and Computer Engineering, Virginia Tech, Blacksburg, VA, USA. [3]Grado Department of Industrial and Systems Engineering, Virginia Tech, Blacksburg, VA, USA. [4]Department of Bio-medical Engineering, Washington University in Saint Louis, Saint Louis, MO, USA. [5]The Department of Industrial and Manufacturing Engineering, Florida A&M University-Florida State University College of Engineering, Tallahassee, FL, USA. ✉e-mail: zkong@vt.edu

simulation tools[1]. However, owing to the layer-wise melting and solidification, a deposited layer will be re-melted and reheated due to the deposition of the following layers. This phenomenon results in a microstructure change[2,3]. The analysis of this effect currently relies on numerical models, such as the finite element method (FEA) and computational fluid dynamics (CFD), due to the lack of direct thermal measurement methods[4,5]. However, numerical models suffer from high computational costs and inaccuracy due to oversimplified assumptions and boundary conditions.

The extreme shape-forming mechanisms primarily cause the challenge in the direct thermal measurement of deposited material during L-PBF. The melting and solidification speeds of L-PBF are rapid, given the laser typically travels at around 1000 mm/s. As a result, the material's thermal gradient and cooling rate are high. According to the literature, the size of the melt pool can be as small as 100 μm and can solidify within 1 ms[6]. Therefore, at least 100 μm/pixel spatial resolution and 10 kHz frame rate are needed to observe the near melt pool heat-affected zone. A High-speed IR camera with a high-magnification lens may meet these specifications. Still, it can only measure the surface thermal profile instead of the interior of the material (sub-surface)[6,7].

On the other hand, the thermal couple array can be potentially used for sub-surface measurement. However, its bulky size limits its sensitivity and spatial resolution[8]. Moreover, embedding the thermal couple array weakens the part's mechanical strength. In addition, the near-melt pool area has an extreme temperature, easily over 600 °C, especially for high-melting-point metals such as stainless steels and Titanium alloys. Such a harsh environment challenges the sensor's survivability.

In short, a high-spatial and temporal resolution sub-surface thermal profile technique that can survive under high temperatures does not exist. This deficiency hinders research on the fundamental physics of laser powder bed fusion and other processes involving high thermal gradients.

Due to their miniature footprint and high sensitivity, fiber optic sensors have recently drawn attention in L-PBF in-situ monitoring[9–12]. When using fiber optic sensing, lights with a known spectrum are emitted to the fiber, and by analyzing the reflection or refraction spectrum, measurements such as strain, bending, vibrations, and temperature can be obtained. Optical fibers typically only have 100–300 μm in diameter; thus, if embedded properly, the damage caused to the measured part is minimal. Moreover, the sensitivity of this type of sensor is high due to its miniature size and its use of optical signals. Fiber-optic sensors have two major types, namely, fully-distributed sensors, such as optical frequency domain reflectometry (OFDR), and point sensors, such as fiber Bragg gratings (FBGs)[10,12].

Hyer and Petrie embedded OFDR into an L-PBF manufactured part to measure the strain during the printing[13]. Similarly, Zou integrated the OFDR into an LPBF part to measure the strain under the load[11]. Hehr et al. attached the OFDR to the backside of an L-PBF substrate to monitor the delamination and cracking near the build-plate surface. Practically, the spatial resolution of OFDR is only a few millimeters due to the limited implementable modulation bandwidth and the trade-off between the number of sampling points and sensing frequency[14].

FBGs are one of the most common fiber optic sensor types. Havermann used FBGs to measure the residual stress-related strain of an L-PBF part[10]. Lerner et al.[9] embedded a fiber with three FBGs into the L-PBF part and measured the temperature evolution during the printing. In their work, the length of the FBGs is 3 mm each with a 30 mm spacing in between.

Figure 1a (top portion) illustrates that FBGs are groups of points that periodically modulate the refractive index along an optical fiber's light-guiding core, allowing light at only specific wavelengths to be reflected[15]. Each group is considered an individual sensor. Figure 1 (bottom portion) demonstrates the reflection spectrum of the sensor on the top. As it shows, it contains four peaks at different wavelengths because four sensors along the fiber have different pitch lengths and reflect the input broadband light at different wavelengths. The spectrum will shift when a specific sensor is subject to temperature change due to thermal expansion and the thermo-optic effect. Such a shift can be used to measure the temperature change. For more details on FBGs, please refer to ref. 15. Due to the weak reflection amplitude of each dot, a complete FBG sensor typically contains thousands of dots to ensure the reflection peak is traceable. However, this level of dot quantity limits the spatial resolution of FBGs to a millimeter level, which is insufficient for near-melt pool area monitoring. Without the capability to measure sharp thermal gradients near the laser-melting zone, the microstructure evolvement in LPBF can not be fully understood.

To significantly improve the spatial resolution of state-of-the-art fiber-optic sensing, this work proposes to use a special type of FBGs called chirped-FBG (C-FBG). As demonstrated in Fig. 1b, a C-FBG has linear chirped periods along the fiber, resulting in different Bragg wavelengths at various locations[15]. Therefore, the spatial information is encoded in the C-FBG reflection spectrum. Thus, an intra-FBG profile can be extracted, enabling much higher spatial resolution than the traditional serial wavelength-division multiplexed FBGs[16,17].

To ensure the C-FBG's survivability under high-temperature conditions in AM processes, the femtosecond laser point-by-point (fs-PbP) method is selected to inscribe the C-FBG[18]. FBG written by the fs-PbP method has been demonstrated to operate up to 1000 °C. However, one drawback of fs-PbP C-FBG is that its reflection spectrum shape exhibits more ripples (see Fig. 1c), which makes it challenging to use the existing model-based demodulation method as typically performed in C-FBG written by UV exposure and a phase mask methods[17,19,20]. To tackle the above challenge, this work proposes to use a machine learning-assisted approach to decode the complicated fs-PbP C-FBG spectrum.

Machine learning (ML) has recently been applied to demodulate traditional uniform FBGs. Zhao et al.[21] used a convolutional neural network (CNN) to extract the effective information of some complex signals in fiber sensing and demonstrated the feasibility of using CNN for demodulation. Djurhuus et al. implemented a Gaussian process regression approach to demodulate FBGs, proving that the result is more accurate than the traditional method[22]. Sarkar et al.[23] demonstrated that ML can discriminate between the strain and temperature effects on FBGs. Li et al. introduced generative adversarial networks (GAN) and dense neural networks (DNN) to demodulate FBGs sensors[24]. Specifically, GAN was used for data augmentation, and DNN was used for wavelength interrogation. Jiang et al.[25] presented that extreme learning machines can improve the demodulation accuracy when the signal of multiple FBGs on a single fiber overlaps. Similarly, Manie et al.[26] used deep learning to improve the accuracy. Kokhanovskiy et al. adopted deep neural networks (DNNs) to demodulate the complex reflectance spectrum caused by densely inscribed FBGs. This work achieved a 1 mm spatial resolution but is still insufficient for closed-to-melt pool area measurement. Using ML on C-FBG for intra-FBG measurement has not been reported. Intra-FBG means position-dependent information within the sensor can be obtained, transforming single-point measurement to array measurement.

Moreover, embedding fiber-optic sensors in L-PBF conditions is also challenging, as optical fibers are small and fragile. Havermann and Zou electroplate the fiber with copper for better protection[10,11]. The electroplated region was melted and bounded with the surrounding material during the embedding. This method only works for isothermal strain measurement. The fiber must stay in a strain-free environment for an accurate thermal measurement. In other words, the fiber cannot be bound to the substrate or part. To achieve the strain-free condition, Lerner et al.[9] micro-welded a metal tube into the workpiece and then fed the fiber through it for temperature measurement. In this process, a grove with a 500 μm diameter is first printed on the workpiece, and then the workpiece is removed for

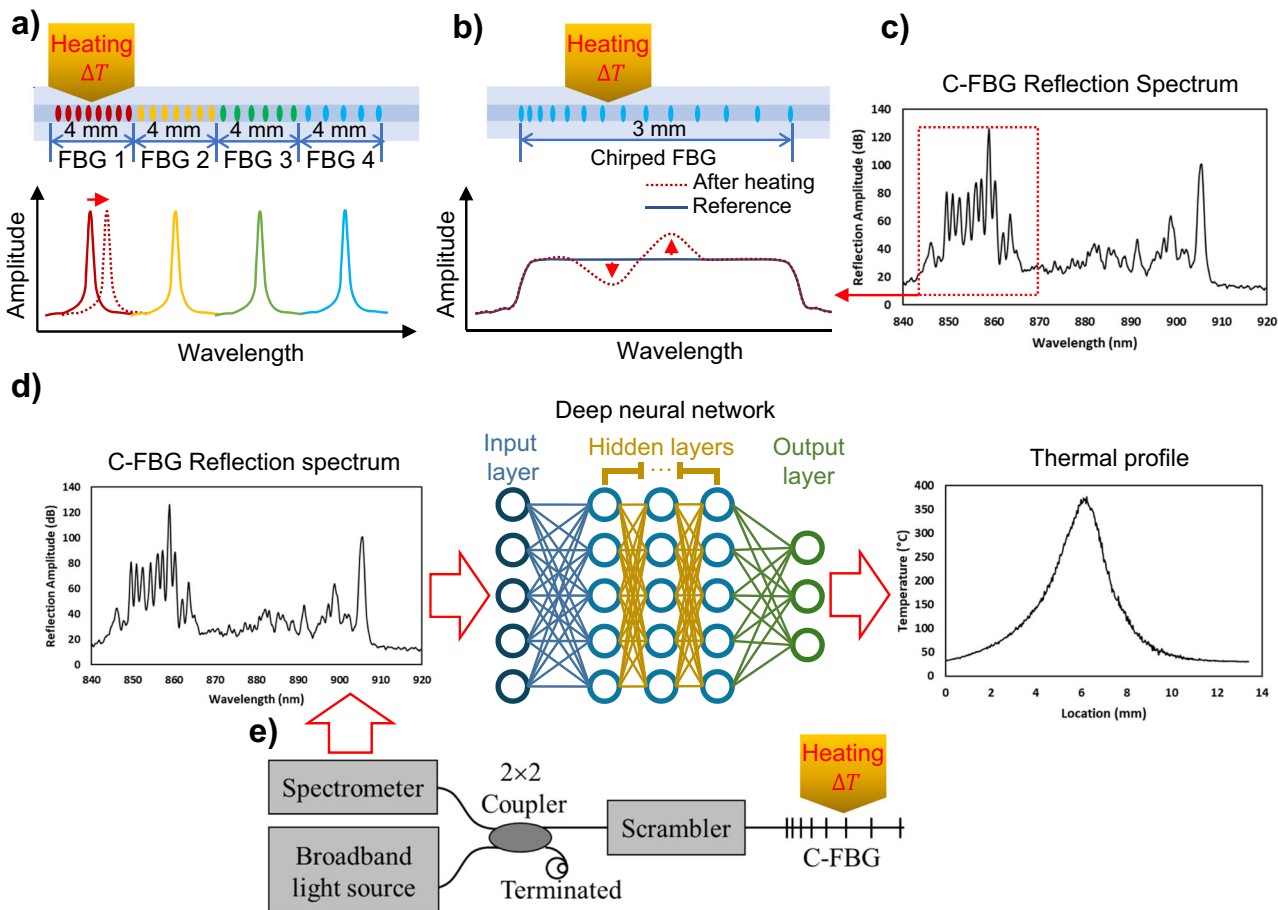

**Fig. 1 | C-FBG background and proposed system illustration. a** The illustration of the FBG working principle, **b** The illustration of the C-FBG working principle, **c** An experimental complex C-FBG reflection spectrum, **d** An illustration of the ML-assisted demodulation method, and **e** The C-FBG data acquisition system.

cleaning, tube positioning, and micro-welding. After that, the work-piece is placed back into the chamber, and the machine directly prints the rest onto the top surface. This method would potentially damage the recoating system as the top half of the tube (200 μm in diameter) is significantly higher than a typical L-PBF layer height (20–100 μm). In addition, such metal tubes are expensive and hard to fabricate. Therefore, in this work, an easy-to-implement fiber embedding method is also introduced.

In summary, though there is a strong need for studying the reheating of L-PBF, accurate and direct measurement does not exist due to challenges such as high spatial-temporal resolution requirement, complex signal demodulation, sensor survivability, and missing an effective embedding technique.

In this work, C-FBG is used as the candidate to improve the spatial resolution of the measurement without sacrificing the sensing frequency. An ML-assisted demodulation model is developed to convert optical signals to thermal profiles. A fiber embedding technique is proposed to couple fiber optical sensors to additively manufactured parts with high precision. The proposed sensor and embedding method is deployed in an L-PBF machine and achieves subsurface thermal measurement during the printing. Information such as thermal gradient and cooling rate can be extracted. The developed sensor provides a tool for studying the reheating and re-melting in L-PBF, enabling future microstructure control.

## Results and discussion
As explained in the *Introduction* section and illustrated in Fig. 1c, the existing methods cannot demodulate the fs-PbP C-FBG reflection spectrum due to the complexity of the signals. This work proposes an

ML data-driven approach to address this issue. In Fig. 1d, the trained neural network takes the C-FBG reflection spectrum (dimension: 1×800) as input and outputs the thermal profile (dimension: 1×480) with 28.8 μm/pixel spatial resolution. The C-FBG reflection spectrum is collected by a data acquisition system (Fig. 1e). The details are provided in the "Method" section. In this ML-assisted sensing system, the spatial resolution is only limited by the IR camera's spatial resolution. This is because when using the trained model for demodulation, the output will always retain the same format and share the same physical meaning and resolution as the thermal profile collected by the IR camera. Note that the spectrometer has a higher spatial resolution than the IR camera. Therefore, it will not become the bottleneck. The training and testing data of the ML model is acquired by a special procedure on a customized calibration system.

### Calibration system
The calibration system aims to create different thermal profiles on the C-FBG, enabling synchronized IR and spectrometer measurement. As shown in Fig. 2a, this system contains a calibration platform that can create a controlled thermal profile on the C-FBG, a direct current (DC) power supply to power the heating hot wire, and monitors that can observe the positioning and thermal information of the C-FBG.

Figure 2b illustrates the components of the calibration platform in Fig. 2a. A fiber holder is used to mount the optical fiber and is fixed to the bottom breadboard. An IR camera (Micro-epsilon, thermoIMAGER TIM VGA) monitors the thermal profile along the fiber during the experiments. The testing stage has two stainless steel blocks with round top surfaces and notches. They are used for mounting the hot wire (36 GA Nichrome 60 Round Resistance Wire), which is heated by

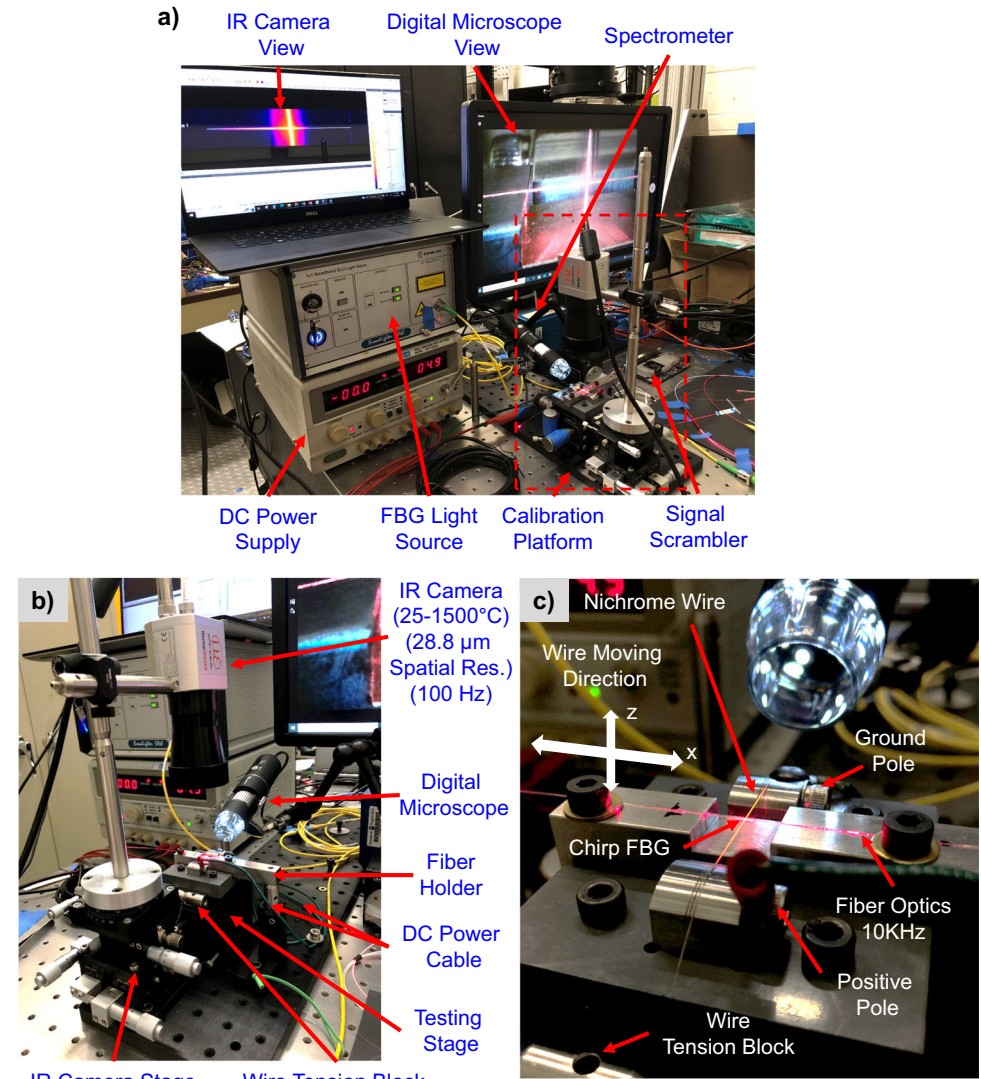

**Fig. 2 | Calibration system for ML-assisted fiber-optic sensing. a** Calibration system main component, **b** Calibration platform, and **c** Calibration platform zoom-in view.

DC. These two metal blocks also serve as DC poles. They are connected to a DC power supply, which controls the hot wire temperature by voltage.

As shown in Fig. 2c, the Nichrome hot wire is positioned below the fiber and mounted onto two metal blocks. These two blocks are attached to a plastic holder as an electrical insulator and structure support. The plastic holder is then attached to a high-precision translational stage (1 μm resolution, ThorLabs MBT616D), which moves the hot wire in both the fiber direction (X) and vertical direction (Z) to create different heating locations and concentrations on the C-FBG. The hot wire is fixed by a screw on the ground pole side block, hanging freely on the other end. The free end is connected to a wire tension block. This block uses its weight to pull the hot wire, keeping it straight under the thermal expansion. A digital microscope ensures the distance between the fiber optic and the hotwire.

**Calibration data collection**

In this work, 14 calibration experiments were performed with various conditions, referred to as cases. During these experiments, synchronized data (IR and spectrometer) was collected. Among all the cases, thirteen were used as training, and one was left for testing (the selection will be explained). The variation was achieved by altering the location and temperature of the hotwire. As a result, thermal profiles

with different temperatures (ranging from 23 to 800 °C) and distribution can be obtained, improving the model's accuracy. These 14 experimental cases can be separated into four categories: horizontal moving, distance cycle, thermal cycle, and reference. The illustration is provided in Fig. 3a–c. In horizontal moving cases (Fig. 3a), the hotwire was positioned 50 μm under the C-FBG. Then, the hot wire moved from 0.5 mm left of the C-FBG to 0.5 mm right. The hot wire temperature (controlled by DC voltage) remained constant for each case. In total, 11 horizontal moving cases were tested. The corresponding power supply voltages were 1.5 V to 6.0 V with 0.5 V incensement (used for model training) and 5.3 V (used for model testing).

To simulate different thermal profile distributions, a distance cycle case (Fig. 3b) was conducted. In this case, the wire was first positioned 50 μm underneath the center of the C-FBG with 5.5 V DC heating. The hotwire then moved 4 mm away from the C-FBG and moved back. When the hot wire was closed to the C-FBG, the thermal distribution sharply peaked at the center. When it was far away, the peak was flattened. To cover additional conditions, a thermal cycle case (Fig. 3c) was conducted as well. In this case, the hot wire was positioned 50 μm underneath the center of the C-FBG, and no movement was involved. Instead, the DC power supply was initially set to 5.5 V, then dropped to zero, followed by ramping back. Besides all these datasets, a reference case was also collected with no heating and

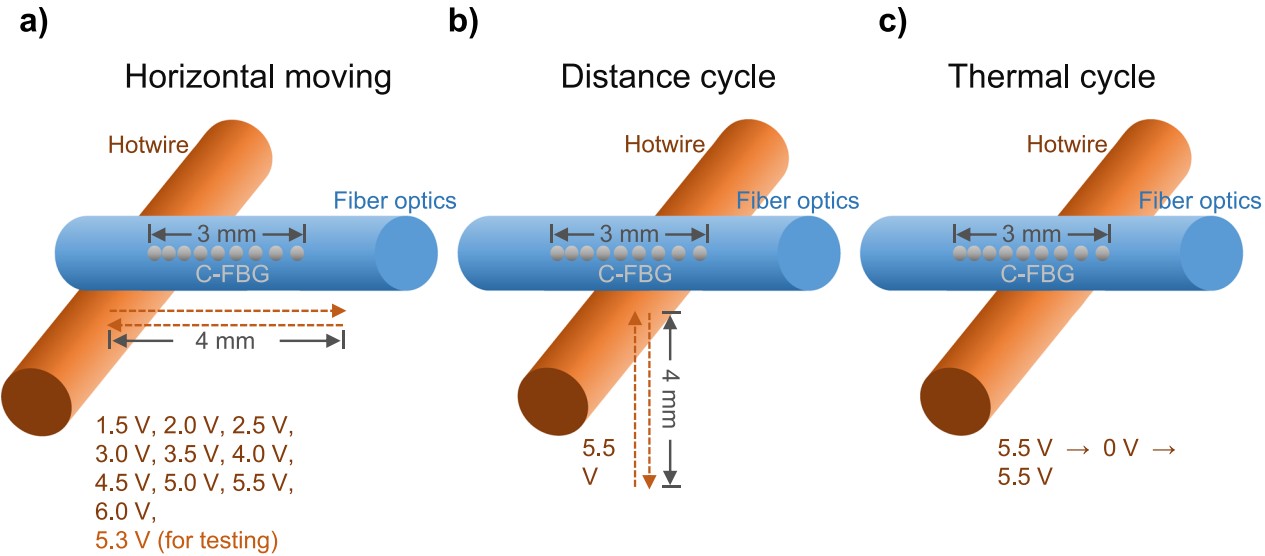

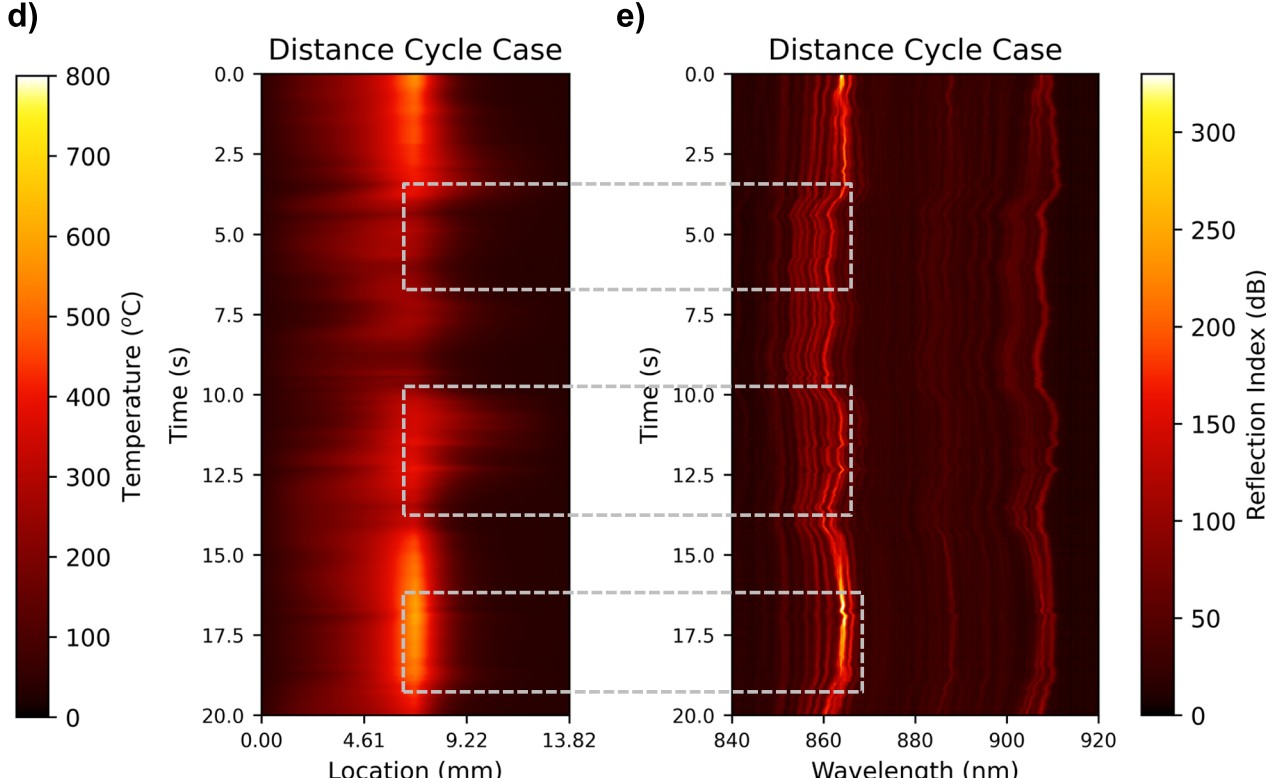

**Fig. 3 | Illustration of calibration procedures and example calibration data.**
**a** Experimental procedure illustration for horizontal moving cases, **b** Experimental procedure illustration for distance cycle case, and **c** Experimental procedure illustration for thermal cycle case. **d** Distance cycle case in-situ thermal profile (IR) dataset visualization, **e** Distance cycle case in-situ reflection spectrum (C-FBG) dataset visualization. Source data are provided as a Source Data file.

movement involved. All the datasets were collected at 100 Hz (80 μs exposure time) with 20 seconds length.

The hot wire's moving speed, both in the horizontal and vertical directions, was ~0.25 mm/s, and the voltage ramping speed was ~0.35 V/s. However, they were not precisely controlled in this study. This is because the calibration of the C-FBG depends on the synchronized IR data that continuously monitors the temperature profile along the fiber. The position and temperature of the hot wire were not used for calibration. Since the fiber optics has a small size, we assume that its temperature reaches a steady state faster than the 80 μs exposure time. During the experiment, the thermal distribution fluctuated due

to ambient air movement, which benefited the training since the covered distributions were more diverse. Except for the 5.3 V horizontal moving case (for testing), all the others were used for model training. The visualization of the C-FBG reflection spectrum (training input data) and the thermal profile (training output data) are provided in Supplementary Figures. They are Supplementary Figs. 1–13 and Supplementary Figs. 14–26, respectively.

An example of thermal profile and C-FBG training datasets is visualized in Fig. 3d, e, respectively. In Fig. 3d, the horizontal axis represents the location, and the vertical axis represents the time. The color indicates the temperature, whereas dark red represents 800 °C.

From the location (horizontal axis) point of view, the hottest temperature appears at the center, which matches the experimental condition. Some fluctuations were overserved due to the air circulation of the room disturbing the surrounding airflow near the fiber. From the time (vertical axis) point of view, the temperature decreases and then ramps back up as designed. Figure 3e is the C-FBG data of the distance cycle case, where the horizontal and vertical axes represent wavelength and time, respectively. Again, wavelengths have corresponding locations on the C-FBG. By visually inspecting the regions cropped by the gray dashes, the shape of the C-FBG data strongly correlates with the IR data, demonstrating a strong potential to use machine learning for demodulation.

**Model tuning and performance**

A fully connected Neural Network (NN) of input dimension 800 (based on a spectrometer) and output dimension 480 (based on IR) is used to model the data. Before training, the input and the output data are normalized, i.e., subtracted by the sample mean and then divided by the sample standard deviation. The loss function is defined by the Mean Squared Error (MSE) of NN prediction with the Rectified Linear

Unit (ReLU) activation function. The NN parameters are optimized using Stochastic gradient descent (SGD) with a learning rate of $8 \times 10^{-2}$ on the training dataset. The tuning of the number of hidden layers and batch size involved a systematic search across a grid of values. As a result, three hidden layers were found accurate enough with a steady decrease in dimension to 700, 600, and 500. A batch size of 50 provides the best performance. The search results for the best number of hidden layers and batch sizes are provided in Fig. 4a, b, respectively. Performance indicators are calculated with the Numpy 1.21.5 package in Python 3. In these two figures and the rest of this work, intercept over union (IOU) is used as the primary model performance indicator since the main objective of the model is to measure the thermal profile, not the temperature, at each location. The meaning of the full region of interest (ROI) and the CFBG ROI in these two figures is explained in the next paragraph. The NN training takes around 10 min on NVIDIA 2080 Ti GPU with Python 3 and PyTorch 1.9.0 for 5000 training iterations. The loss function converged to order $1 \times 10^{-3}$ on the normalized IR data. The training loss convergence graph is provided in Fig. 4c. The average correlation of the testing is 0.996, and the correlation plot is provided in Fig. 4d. This plot shows that all the data points are closely

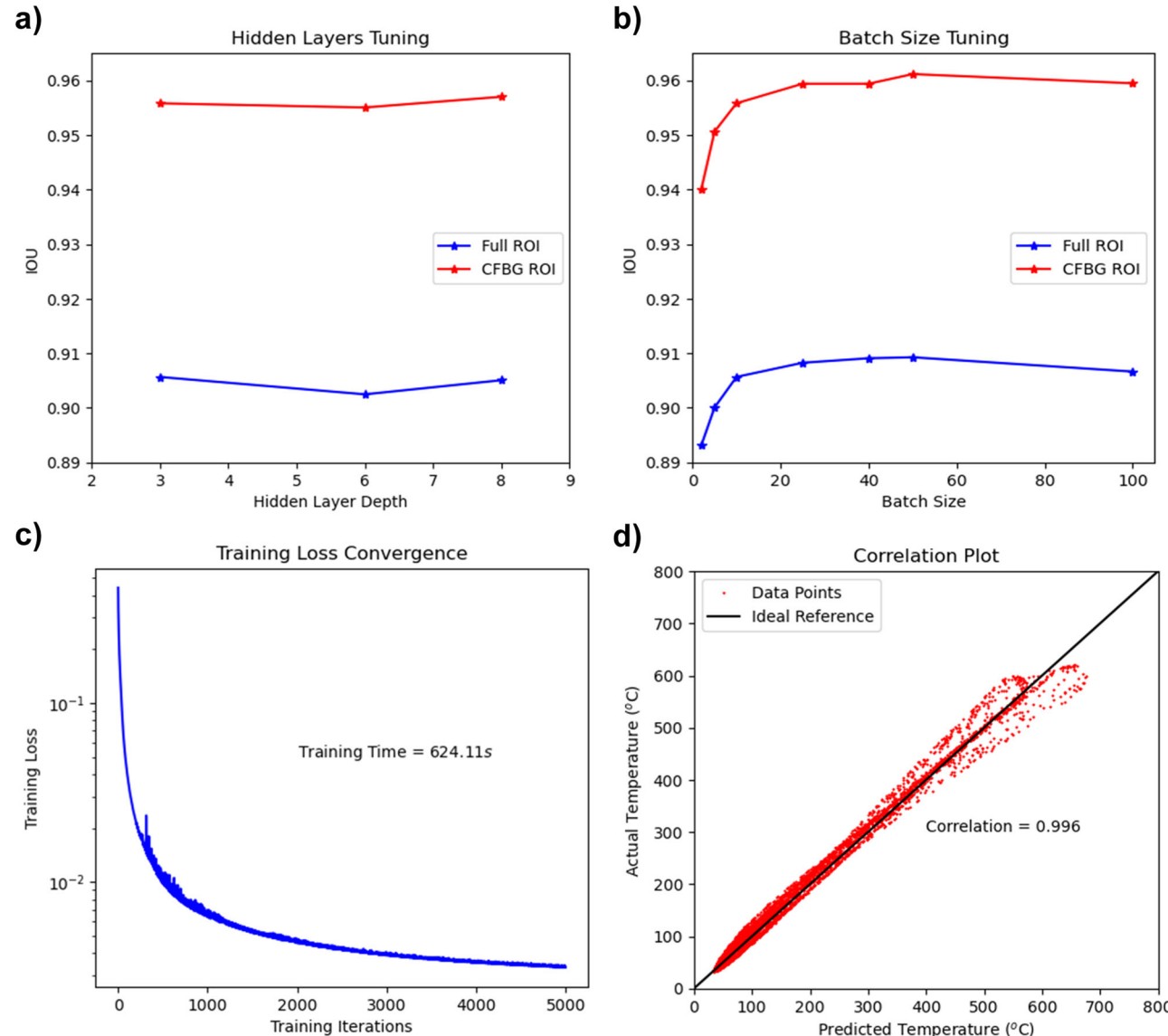

**Fig. 4 | Model training-related performance plots. a** Hidden layer determination, **b** Batch size determination, **c** Training loss convergence, and **d** Correlation plot for the testing case. Source data are provided as a Source Data file.

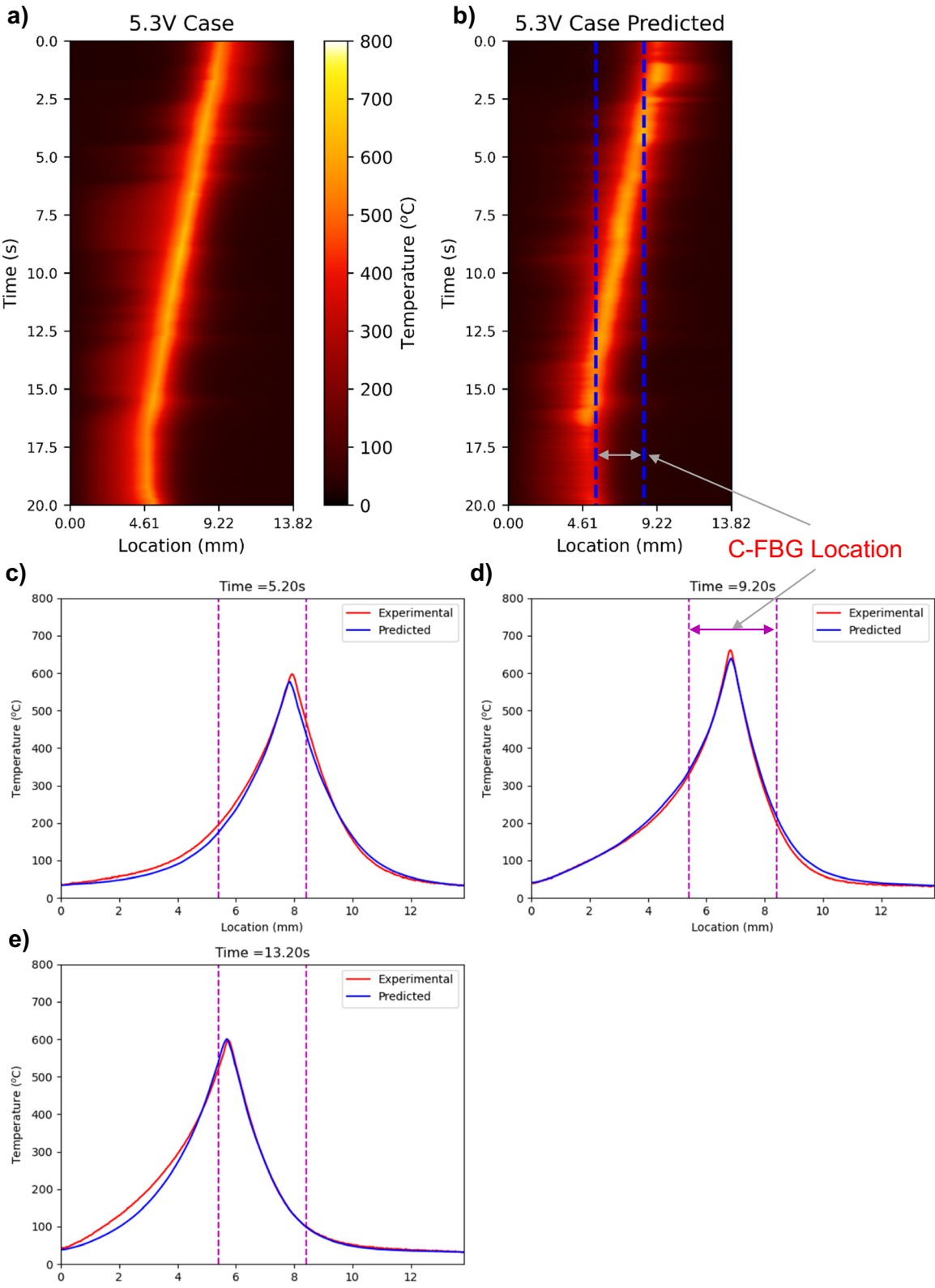

**Fig. 5 | Illustration of AI-demodulation results. a** Experimental 5.3 V horizontal moving IR dataset, **b** 5.3 V case IR ML demodulation result based on experimental C-FBG testing data, **c–e** Examples of experimental and demodulated thermal profiles at various time steps. Source data are provided as a Source Data file.

distributed along the 45-degree reference line, indicating a strong correlation between the experimental and ML-demodulated data.

As mentioned in *Calibration data collection*, the 5.3 V DC horizontal moving case is used to test the trained model's performance. The visualizations of the experimental thermal profile and the AI-demodulated thermal profile are presented in Fig. 5a, b, respectively. Examples of thermal profiles for selected time steps are shown in Fig. 5c–e, where heating location change

**Table 1 | Model performance statistics**

| ROI | Training size | Training loss | Average correlation | Training iterations | Relative error | Max absolute error (°C) | Mean absolute error (°C) | IOU |
|---|---|---|---|---|---|---|---|---|
| FULL Range | 26000 | 0.003 | 0.996 | 5000 | 0.089 | 43.443 | 15.289 | 0.915 |
| C-FBG Range | | | | | 0.041 | 39.976 | 12.708 | 0.967 |

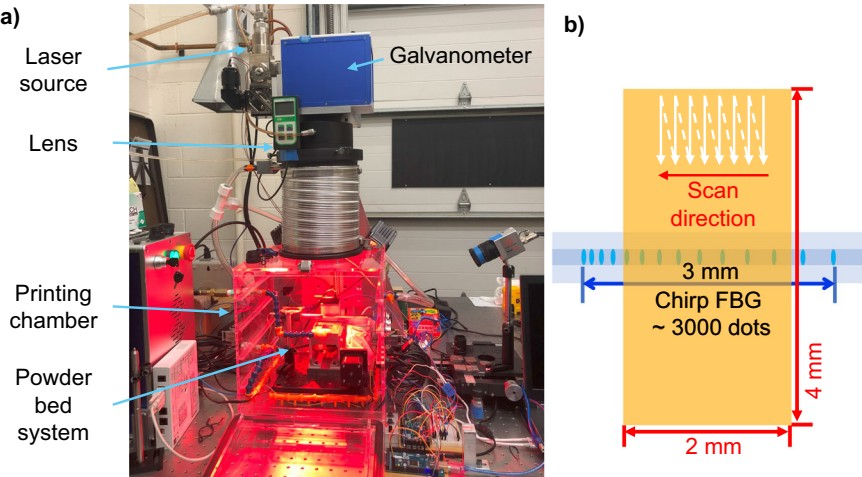

**Fig. 6 | Fiber optics embedding and testing machine and scan pattern. a** The customized multi-sensing L-PBF machine[27], **b** L-PBF melting pattern illustration.

can be clearly observed. As these three figures show, the demodulated thermal profile nicely matched the experimental IR-collected data, especially in the center region (between two dashes). This is because the C-FBG only occupies a portion of the IR camera field of view, making thermal demodulation of the outer region less accurate. There are data outside of the C-FBG covered range because the machine learning demodulation was based on vector (spectrum intensity) to vector (thermal profile) correlation, not point (specific wavelength) to point (specific location) correlation. Thermal profile, in other words, temperature distribution, is a continuous curve. Therefore, the spectrum data generated from the C-FBG also contains the thermal information adjacent to it. As mentioned in the last paragraph, the full range of locations refers to full ROI in this work, and C-FBG ROI refers to the length that C-FBG spans. The same phenomenon can also be observed by comparing Fig. 5a, b. Even though the outer C-FBG regions have less accuracy, they still contain useful information. Therefore, when estimating the model accuracy, the full ROI and C-FBG ROI are analyzed separately.

Table 1 logs all the model performance statistics. As it shows, the IOU within the C-FBG ROI achieves 0.967, a very high performance. The relative error of the C-FBG ROI is 0.041, and the mean absolute error is 12.708 °C, which is negligible for measuring high temperatures (above 500 °C). The maximum absolute error of the C-FBG ROI is 39.976 °C, reflecting the range of the outlier predictions. By examining the data, this type of outlier only happens a few times and will not affect the main purpose of the proposed sensor, which is to measure the cooling rate and the thermal gradient of the melt pool's surrounding area.

**L-PBF sub-surface measurement**

To test the performance of the proposed fiber-optic sensing, it is deployed in a customized multi-sensing L-PBF testing platform developed in our previous work (Fig. 6a)[27]. Figure 6b illustrates that the laser melts a $2 \times 4$ mm² region on the substrate with 800 mm/s speed

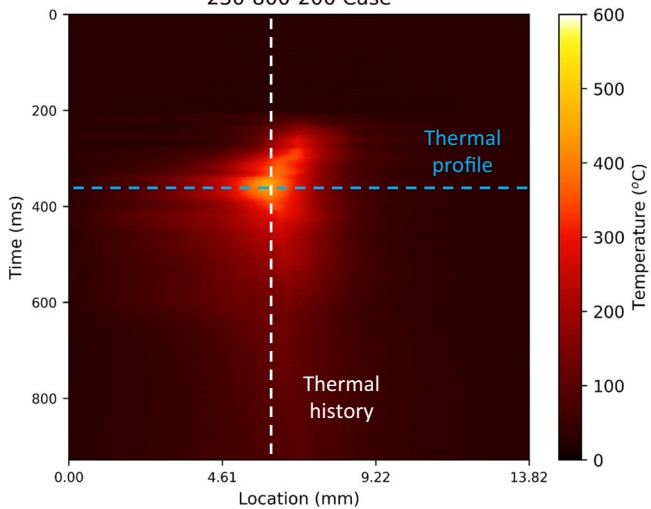

**Fig. 7 | ML-demodulated in-situ sub-surface thermal profile data during the L-PBF process.** Source data are provided as a Source Data file.

and 200 W power in an 80 μm spaced one-directional raster pattern. The C-FBG is located 230 μm underneath the melting surface to collect the in-situ data at 10 kHz. Although this sensing frequency is higher than the calibration cases, the exposure time of each frame remains at 80 μs. Therefore, the calibration is still valid.

The result of the sub-surface thermal measurement is visualized in Fig. 7. In this figure, the data along the white dashed line represents the thermal history of a given location, and the data along the blue dashed line represents the thermal profile along the fiber at a given time step.

Examples of thermal profiles are shown in Fig. 8a–d. At $t = 300$ ms, the laser raster lines reach the center of the C-FBG, creating a thermal profile peaked around 360 °C (Fig. 8a). After 55 ms, the rectangular-

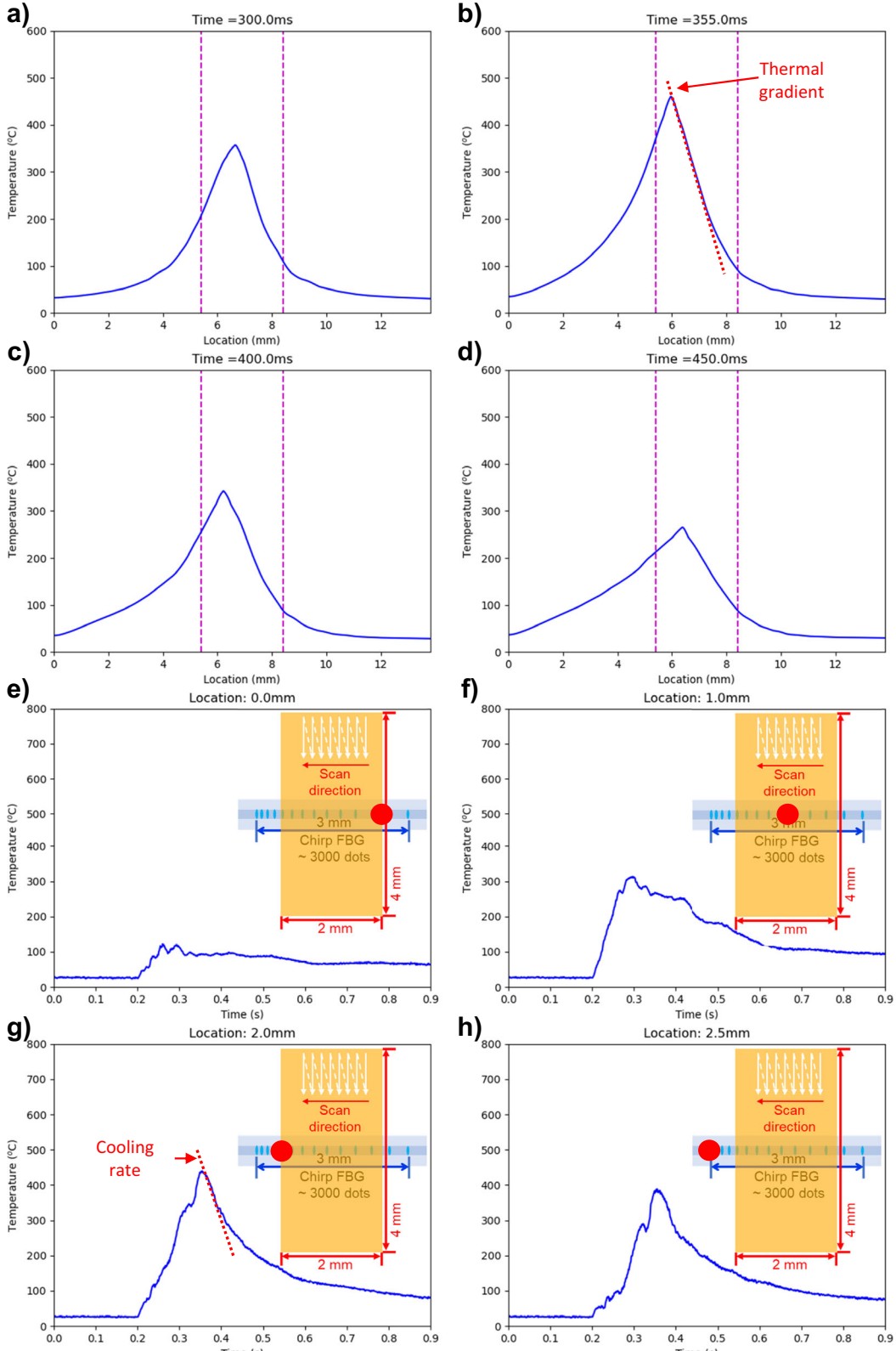

**Fig. 8 | Example measurement of thermal profiles and thermal histories using the proposed sensor. a–d** Selected thermal profile at different time steps. **e–h** Selected thermal history at different locations where the red dot indicates the sampling location along the C-FBG. Source data are provided as a Source Data file.

shaped melting is about to finish, and the laser scanning location moves to the left of the C-FBG, resulting in the peak temperature location also moving to the left (Fig. 8b). Due to the heat accumulation, the peak temperature increases to ~450 °C. This figure demonstrates

that the thermal gradient is close to a linear form (see the red dashed line in Fig. 8b) and can be easily extracted. The sharpest thermal gradient of this monitoring case appears by the end of the scan since it has the highest peak temperature due to heat accumulation, and the value

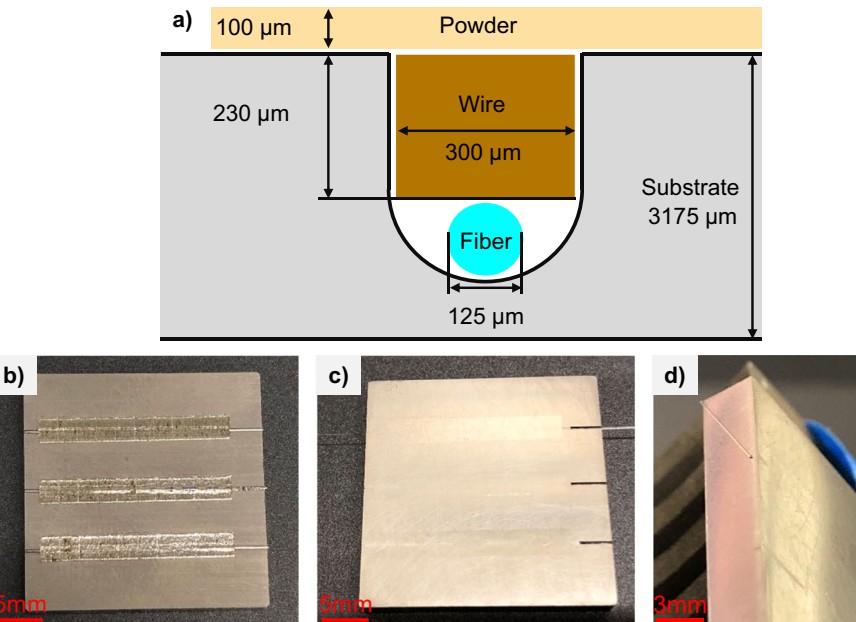

**Fig. 9 | Fiber embedding procedure and results. a** Fiber embedding procedure illustration, **b** Substrate after laser melting, **c** Substrate after polishing, and **d** Substrate with a fiber-optic sensor embedded in.

is $4 \times 10^5\,°C/m$, about one order of magnitude smaller than the melt pool region[28]. This also implies that the thermal gradient within 0.3 mm (about ten layers) from the surface is 1–10 times smaller than the melt pool region. Figure 8c, d show the thermal profiles during the cooling.

Figure 8e–h demonstrate the thermal histories at selected locations. The sharpest cooling rate also appears at the end of the laser scan due to the heat accumulation (Fig. 8g). The value is around 3500 °C/s, around three orders of magnitude smaller than the melt pool region[28]. Interestingly, comparing Fig. 8h, e, the peak temperature and cooling rate at the left end of the C-FBG are significantly higher than at the beginning of the scan, even though no laser melting is directly above. This indicates that sub-surface reheating highly depends on the heat accumulation of the surface melting, and the affected region can be complicated if hatch angle rotation (raster angle change between layers) is used during the printing.

## Summary and future works

This work proposed a type of sensor that integrates ML and fiber-optic sensing techniques. The developed sensor achieved 28.8 μm/pixel spatial resolution and 10 kHz sampling frequency, ideal for measuring sharp thermal gradient and cooling rates in the L-PBF process. A calibration testbed and calibration procedure are developed to train the ML-assisted demodulation model. A minimal-damage embedding technique is also developed to ensure a closed-contact but strain-free environment for the fiber. The case study section demonstrates that the proposed sensor can measure crucial information, such as sub-surface cooling rate and thermal gradient during the L-PBF process, providing a way to study the reheating effect associated with the L-PBF. In the future, more testing will be conducted to cover different process parameter combinations, depths, and materials. Future collected data can be used to calibrate existing FEA and CFD models and enable precise microstructure control in L-PBF. The proposed sensor can also be used for other advanced measurements that require high spatial resolution and sensing frequency. For example, to measure the space shuttle thermal tile's insulation performance and the thermal profile across the Tokamak fusion reactor wall.

## Methods

### C-FBG fabrication and data acquisition system

The fabrication method and setup of the C-FBG follow the authors' previous work[29]. In short, a femtosecond laser pulse with a fixed repetition rate is tightly focused into the light-guiding core of an optical fiber (SMF28, Corning) to induce refractive index modulation. A C-FBG is formed by translating the optical fiber along its axial direction with a linearly increasing speed. In this work, a three mm-long 5th-order C-FBG centered at 855 nm with a chirp rate of 3.33 nm/cm was fabricated. After the C-FBG encrypting, the acrylate fiber coating was removed by acetone to reduce the diameter of the fiber. This step is also crucial for high-temperature sensing because the fiber coating could melt and burn above 300 °C.

As illustrated in Fig. 1e, the data acquisition system uses a broadband light source (S840, Superlum) to launch the light into the C-FBG through a 2×2 fiber coupler (TW805R5F2, Thorlabs). An in-line high-speed polarization scrambler (NOPS-110210131, Agiltron) was implemented to minimize the variation of the C-FBG spectrum induced by polarization perturbation during the measurement. The reflected light from the C-FBG is routed by the same 2×2 coupler and detected by a high-speed spectrometer (Max sampling rate: 70 kHz, C-00116, Wasatch Photonics) for spectrum analysis. This spectrometer uses a high-efficiency volume phase holographic (VPH) grating to disperse the input light onto a high-speed one-dimensional camera, where individual wavelengths are encoded by different pixels and acquired simultaneously. The spectrometer communicates with the host computer via CameraLink protocol, allowing full spectrum acquisition at up to 70 kHz speed.

### Fiber embedding procedure

This work also develops a fiber embedding procedure to ensure minimal damage to the part and a tight fiber fit. Figure 9a illustrates that a slot with 300 μm width and 355 μm depth is first etched on the Ti-64 substrate by a wire EDM machine to create a miniature hole. Then, a precisely ground rectangular cross-section Ti-64 wire fills the gap between a plain fiber and substrate surface. After that, a layer of 100-μm-thick Ti-64 powder is applied on the substrate surface and melted by a customized laser powder bed fusion machine. The melting result is demonstrated in Fig. 9b. After that, the plain fiber is retracted

and examined to ensure a strain-free condition is achieved, and then, the substrate's surface is polished, as shown in Fig. 9c. At this point, the C-FBG encrypted fiber sensor can be inserted. Figure 9d provides a detailed view of the embedded result with a fiber-optic sensor passing through under a strain-free condition. The depth of the slot and thickness of the wire can be altered to study the thermal history at different depths.

## Reporting summary

Further information on research design is available in the Nature Portfolio Reporting Summary linked to this article.

## Data availability

The data used to generate all the main and Supplementary Figs. and are available in Figshare under the accession code [https://doi.org/10.6084/m9.figshare.25036454].

## Code availability

The raw data required to reproduce these findings and the code are available in Figshare under the accession code [https://doi.org/10.6084/m9.figshare.25036454].

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

## Acknowledgements

We would like to thank Scott Lancaster, Randall Waldron, Kelly Snidow, Joe Linkous, and Shawn Culver from Virginia Tech for assisting with platform setup and substrate fabrication. The research reported in this publication was supported by the Office of Naval Research under Award Number N00014-18-1-2794.

## Author contributions

Rongxuan W. and Z.J.K. proposed the overall solution. Rongxuan W. designed and fabricated the experimental platform. Ruixuan W. designed and fabricated the C-FBG and instrumented the interrogation system. Rongxuan W., Ruixuan W., and C.D. participated in the experiments. Rongxuan W. & R.G. performed the ML model training and data analysis. Rongxuan W., S.Y., R.G., and Z.J.K. wrote the main text of the paper. A.W. and Z.J.K. provide the resources and funding acquisition. All authors discussed the results and contributed to the writing of the final manuscript.

## Competing interests

The authors declare no competing interests.
