## [Peer Review File · Nature Communications]

Sub-surface thermal measurement in additive manufacturing via machine learning-enabled high-resolution fiber optic sensingREVIEWER COMMENTS

Reviewer #1 (Remarks to the Author):

Summary:

This work addresses the challenge of studying microstructure evolution in metal parts during additive manufacturing (AM). It introduces a machine learning approach to enhance the spatial resolution of chirped-fiber Bragg grating (C-FBG) sensors to 28.8 μm , from millimeter levels. A sensor embedding technique is also developed to protect parts from heat damage, facilitating precise temperature measurements during AM processes, with potential applications in understanding metal AM physics.

The authors have presented novel research in this manuscript that would be crucial in furthering the fundamental understanding of the LPBF process. The research has been well-presented and thorough work has been done.

I have no additional comments for the authors.

Reviewer #2 (Remarks to the Author):

The main problem with this manuscript is the numerous English language errors. Please have a native English speaker review the manuscript for tenses, articles, and similar common mistakes. examples include "allowing embedded fiber stays" on page 7, and "accurate information is not exist" on page 6.

Section 2.3 (training and testing dataset) is confusing. I think when you say "case" you mean "training data set" or "subset". I would review the terminology to better describe the training data subsets. It seems there are several training data subsets, one that varies horizontal hotwire position, one that varies vertical hotwire position, and one that varies hot wire applied voltage. Please make that more clear and don't use the word "case". Try "training data subset" instead.

Page 15 - maximum absolute error is better than mean absolute error here. In other words, error bars that contain the range of outliers versus the average of the outliers.

Page 16 - after you discuss the novel procedure for embedding the fiber, how do you know it is not bonded to the surface? I'm sure you checked this somehow. Can you simply pull it in and out to see if it slides easily? Please add one sentence to describe how this is checked.

I didn't know when rastering the turning angle is called the "hatch angle". I might be an idiot, but it could help others to include a brief definition - totally optional.

What is the operating principle for the high-speed spectrometer? Does it use an SLM, shutter, rotor, or other high speed method? Might help to include a brief explanation.

All in all a fine paper, and interesting. I enjoyed reading it.

Reviewer #3 (Remarks to the Author):

This paper proposes an interesting method to study the thermal profile during laser powder bed fusion process. In the work, machine learning is being used to demodulate the chirped Fiber bragg grating reflection spectrum. This is done by a calibration platform with a series of dataset (mostly

1D along the fiber grating) to train the thermal profile.

Overall, this paper is organized and presented well, minus point 1 listed below. However, I actually think this paper is a better fit for more technical journals such as optical express, IEEE sensors, etc (the authors cited several papers from these journals), but not necessarily Nature

Communications. My questions:

1. Section 1.2 should be more succinct. It is not necessary to cover too much basic FBG backgrounds like a review paper.

2. In Section 2 methods, the authors list the C-FBG spectrum resolution to be 800 points from 840nm to 920nm. But we know that a typical OSA (optical spectrum analyzer) can scan a much higher resolution, albeit slower. Is it because it being too slow for even calibration (one time measurement)?

Similarly in Figure 5, no detail about moving speed, voltage ramping speed/profile is mentioned. We also don't know how these parameters affect the equilibrium of the thermal dynamics. Are the authors sure that data was taken after wire/fiber reach steady states?

2. In Figure 8(a), 8(b), since the authors acknowledged that "C-FBG only occupies a portion smaller than IR camera field of view", then why is there any data at all outside of the CFBG, where no grating exists thus no reflection signal? If the result is consistent, shouldn't that area after CFBG cutoff be blue instead?

3. I find it confusing that the 28.8 um/pixel spatial resolution is directly coming from the IR camera. Due to the neural network calibration process mentioned in the paper, how did the IR camera spatial resolution translate into the sensors spatial resolution? Why is it 1:1?

Responses to the Reviewers

We sincerely thank the reviewers for the efforts and time in reviewing this work, which has undoubtedly helped strengthen our work. We have addressed the shortcomings pointed out by the reviewers in the rebuttal document and revised the manuscript accordingly.

The model code and dataset are uploaded via Figshare as a zip folder through the submission portal. The zip folder contains a readme.txt file that explains all the files enclosed. According to the message in the submission portal, the shared link will only be available to us after this round of submissions. That link will be added to the Data Availability section before final acceptance. The reviewers can access the data and the code through the submission/review portal. Please let us know if there is any problem.

Response to Reviewer 1

Reviewer 1 Comments

This work addresses the challenge of studying microstructure evolution in metal parts during additive manufacturing (AM). It introduces a machine learning approach to enhance the spatial resolution of chirped-fiber Bragg grating (C-FBG) sensors to 28.8 μm , from millimeter levels. A sensor embedding technique is also developed to protect parts from heat damage, facilitating precise temperature measurements during AM processes, with potential applications in understanding metal AM physics.

The authors have presented novel research in this manuscript that would be crucial in furthering the fundamental understanding of the LPBF process. The research has been well-presented and thorough work has been done.

I have no additional comments for the authors.

Authors' Response:

We sincerely thank the reviewer for the effort and time invested in this work.

Response to Reviewer 2

We sincerely thank the reviewer for the effort and time invested in this work. We have addressed the shortcomings pointed out by the reviewer in the revised manuscript. Enclosed herewith is our response to the reviewer's questions. The revisions are highlighted in yellow in the revised manuscript. A summary of the changes is listed after each question.

(Q1) The main problem with this manuscript is the numerous English language errors. Please have a native English speaker review the manuscript for tenses, articles, and similar common mistakes. examples include "allowing embedded fiber stays" on page 7, and "accurate information is not exist" on page 6 .

Authors' Response:

We truly appreciate the reviewer's concerns regarding grammar. The manuscript language has been improved significantly.

(Q2) Section 2.3 (training and testing dataset) is confusing. I think when you say "case" you mean "training data set" or "subset". I would review the terminology to better describe the training data subsets. It seems there are several training data subsets, one that varies horizontal hotwire position, one that varies vertical hotwire position, and one that varies hot wire applied voltage. Please make that more clear and don't use the word "case". Try "training data subset" instead.

Authors' Response:

We truly appreciate the reviewer's concerns regarding the clarity of terminology, and we agree that clearly defined terminology improves the readability of this manuscript. We used the word 'case' because it implies labeling different experimental conditions. Training or testing datasets are the roles of those cases served for model building. Therefore, we believe explaining our original terminology clearly is the best solution. We heavily modified the first paragraph of Section 2.3 to make the manuscript more clear. All the changes are shown below.

Changes made in the first paragraph of Sec. 2.3 (page 11):

2.3 Experimental Conditions and Datasets

In this work, 14 calibration experiments were performed with various conditions, which are referred to as cases. During these experiments, synchronized data (IR and spectrometer) was collected. Among all the cases, thirteen of them were used as training, and one was left for testing (the selection will be explained). The variation was achieved by altering the location and temperature of the hotwire. As a result, thermal profiles with different temperatures (ranging from 23 to 800 °C) and distribution can be obtained, and this will improve the accuracy of the model.

(Q3) Page 15 - maximum absolute error is better than mean absolute error here. In other words, error bars that contain the range of outliers versus the average of the outliers.

Authors' Response:

We truly appreciate the reviewer's concerns regarding the error. Maximum absolute error has been added, and it is around 40 °C. It is larger than the mean absolute error (around 13°C) but does not happen very frequently and should be considered an outlier. As the primary purpose of the proposed sensor is to measure

the cooling rate and the thermal gradient of the melt pool's surrounding area, the outlier will not significantly affect the result.

Changes made in the last paragraph of Sec. 2.4 (page 16), and Table 1:

Table 1 logs all the model performance statistics. As it shows, the IOU within the C-FBG ROI achieves 0.967, a very high performance. The relative error of the C-FBG ROI is 0.041, and the mean absolute error is 12.708 °C, which is negligible for measuring high temperatures (above 500 °C). The maximum absolute error of the C-FBG ROI is 39.976 °C, reflecting the range of the outlier predictions. By examining the data, this type of outlier only happens a few times and will not affect the main purpose of the proposed sensor, which is to measure the cooling rate and the thermal gradient of the melt pool's surrounding area.

Table 1. Model performance statistics.

ROI	Training Size	Training Loss	Average Correlation	Training iterations	Relative Error	Max Absolute Error (°C)	Mean Absolute Error (°C)	IOU
FULL Range					0.089	43.443	15.289	0.915
C-FBG Range	26000	0.003	0.996	5000	0.041	39.976	12.708	0.967

(Q4) Page 16 - after you discuss the novel procedure for embedding the fiber, how do you know it is not bonded to the surface? I'm sure you checked this somehow. Can you simply pull it in and out to see if it slides easily? Please add one sentence to describe how this is checked.

Authors' Response:

We truly appreciate the reviewer's concerns regarding the strain-free condition of the embedded fiber. To protect the C-FBG encrypted fiber, we use a plain fiber to complete the embedding procedure and replace it with the C-FBG encrypted fiber for measurement experiments. Therefore, the strain-free condition is guaranteed. We added a short description to the manuscript to improve the clarity of our manuscript.

Changes made in the first paragraph of Sec. 2.5 (page 17):

This work also develops a fiber embedding procedure to ensure minimal damage to the part and a tight fiber fit. As Figure 9a illustrates, a slot with 300 μm width and 355 μm depth is first etched on the Ti-64 substrate by a wire EDM machine to create a miniature hole. Then, a precisely ground rectangular cross-section Ti-64 wire fills the gap between a plain fiber and substrate surface. After that, a layer of 100 μm-thick Ti-64 powder is applied on the substrate surface and melted by a customized laser powder bed fusion machine. The melting result is demonstrated in Figure 9b. After that, the plain fiber is retracted and examined to ensure a strain-free condition is achieved, and then, the substrate's surface is polished, as shown in Figure 9c. At this point, the C-FBG encrypted fiber sensor can be inserted.

(Q5) I didn't know when rastering the turning angle is called the "hatch angle". I might be an idiot, but it could help others to include a brief definition - totally optional.

Authors' Response:

We appreciate the reviewer's concern about this terminology. The hatch angle is the angle between the rastering line and the y-axis. A clarification has been added to the manuscript.

Changes made on the last paragraph of Sec. 3.2 (page 21):

This indicates that sub-surface reheating highly depends on the heat accumulation of the surface melting, and the affected region can be complicated if hatch angle rotation (raster angle change between layers) is used during the printing.

(Q6) What is the operating principle for the high-speed spectrometer? Does it use an SLM, shutter, rotor, or other high speed method? Might help to include a brief explanation.

Authors' Response:

We truly appreciate the reviewer's concerns regarding the operating principle for the high-speed spectrometer. The spectrometer is a commercial product from Wasatch Photonics (model number C-00116). It uses a high-efficiency volume phase holographic (VPH) grating to disperse the input light onto a high-speed one-dimensional camera, where individual wavelengths are encoded by different pixels and acquired simultaneously. The spectrometer communicates with the host computer via CameraLink protocol, allowing full spectrum acquisition up to 70kHz speed. This information has been added to the manuscript.

Changes made in the last paragraph of Sec. 2.1 (page 9):

The reflected light from the C-FBG is routed by the same 2×2 coupler and detected by a high-speed spectrometer (Max sampling rate: 70 kHz, C-00116, Wasatch Photonics) for spectrum analysis. This spectrometer uses a high-efficiency volume phase holographic (VPH) grating to disperse the input light onto a high-speed one-dimensional camera, where individual wavelengths are encoded by different pixels and acquired simultaneously. The spectrometer communicates with the host computer via CameraLink protocol, allowing full spectrum acquisition up to 70kHz speed.

Response to Reviewer 3

Reviewer's Overall Comment:

This paper proposes an interesting method to study the thermal profile during laser powder bed fusion process. In the work, machine learning is being used to demodulate the chirped Fiber Bragg grating reflection spectrum. This is done by a calibration platform with a series of dataset (mostly 1D along the fiber grating) to train the thermal profile.

Overall, this paper is organized and presented well, minus point 1 listed below. However, I actually think this paper is a better fit for more technical journals such as optical express, IEEE sensors, etc (the authors cited several papers from these journals), but not necessary Nature Communications.

Authors' Response to Reviewer's Overall Comment:

We sincerely thank the reviewer for the effort and time invested in this work. We appreciate all the comments and suggestions, which have undoubtedly helped strengthen our work. We have tried to address the shortcomings pointed out by the reviewer in the revised manuscript. Attached herewith is our response to the reviewer's questions. The resulting changes to the content are marked in yellow.

We truly appreciate review's suggestion regarding considering other journals. However, we believe this manuscript's scope fits Nature Communications well since fibre optics and optical communications is one of the major subjects that Nature Communications covers (<https://www.nature.com/subjects/fibre-optics-and-optical-communications/ncomms>). The follows are some examples of fiber optics sensing paper published on Nature Communications:

- Chen, C.W., Nguyen, L.V., Wisal, K. et al. Mitigating stimulated Brillouin scattering in multimode fibers with focused output via wavefront shaping. Nat Commun 14, 7343 (2023). <https://doi.org/10.1038/s41467-023-42806-1>
- Li, C., Wieduwilt, T., Wendisch, F.J. et al. Metafiber transforming arbitrarily structured light. Nat Commun 14, 7222 (2023). <https://doi.org/10.1038/s41467-023-43068-7>
- Strutyński, C., Evrard, M., Désévéday, F. et al. 4D Optical fibers based on shape-memory polymers. Nat Commun 14, 6561 (2023). <https://doi.org/10.1038/s41467-023-42355-7>
- Li, J., Zhu, W., Biondi, E. et al. Earthquake focal mechanisms with distributed acoustic sensing. Nat Commun 14, 4181 (2023). <https://doi.org/10.1038/s41467-023-39639-3>
- Kaushal, S., Aadhi, A., Roberge, A. et al. All-fibre phase filters with 1-GHz resolution for high-speed passive optical logic processing. Nat Commun 14, 1808 (2023). <https://doi.org/10.1038/s41467-023-37472-2>
- Badt, N., Katz, O. Real-time holographic lensless micro-endoscopy through flexible fibers via fiber bundle distal holography. Nat Commun 13, 6055 (2022). <https://doi.org/10.1038/s41467-022-33462-y>
- Plöschner, M., Morote, M.M., Dahl, D.S. et al. Spatial tomography of light resolved in time, spectrum, and polarization. Nat Commun 13, 4294 (2022). <https://doi.org/10.1038/s41467-022-31814-2>

(Q1) Section 1.2 should be more succinct. It is not necessary to cover too much basic FBG backgrounds like a review paper.

Authors' Response:

We truly appreciate the reviewer's concerns regarding shortening Section 1.2. We believe it is beneficial to explain the FBG background thoroughly because that helps the audience from different engineering backgrounds to fully understand the intuition and methodology of this work. Following the reviewer's suggestion, we made Section 1.2 more concise and added a sentence that, for more details of FBG, readers can refer to the citation [15].

[15] Erdogan, T., Fiber grating spectra. Journal of lightwave technology, 1997. 15(8): p. 1277-1294.

(Q2) In Section 2 methods, the authors list the C-FBG spectrum resolution to be 800 points from 840nm to 920nm. But we know that a typical OSA (optical spectrum analyzer) can scan a much higher resolution, albeit slower. Is it because it being too slow for even calibration (one time measurement)?

Authors' Response:

We truly appreciate the reviewer's concerns regarding the use of OSA. It is correct that a typical Optical Spectrum Analyzer (OSA) could be employed to achieve a high-resolution spectrum for calibration. However, in this work, we have intentionally chosen to keep consistent spectrum acquisition conditions, such as the equipment used, number of spectrum samples, and exposure time, for both calibration and measurement phases. This approach is designed to minimize potential errors from variations in light response and spectral sampling numbers.

It's worthwhile to highlight that for the Additive Manufacturing (AM) process examined in our study, a measurement speed greater than 10 kHz is required. Consequently, a typical scanning OSA, which might not meet these speed demands, was not suitable for our purposes. Instead, we have utilized a camera-based spectrometer. This type of spectrometer captures the entire spectrum in a single shot, with different pixels recording different spectral components, thereby offering the high acquisition speed necessary for our study, albeit at the cost of reduced spectral resolution.

We acknowledge and appreciate the value of the reviewer's suggestion. In our future work, we plan to explore the potential of integrating high-resolution, low-speed spectrum acquisition during the calibration phase. This may further enhance the accuracy of our model, and we are grateful for your recommendation in this regard.

(Q3) Similarly in Figure 5, no detail about moving speed, voltage ramping speed/profile is mentioned. We also don't know how these parameters affect the equilibrium of the thermal dynamics. Are the authors sure that data was taken after wire/fiber reach steady states?

Authors' Response:

We truly appreciate the reviewer's concerns regarding providing more details about the experimental conditions. The hot wire's moving speed, both in the horizontal and vertical directions, was approximately 0.25 mm/s, and the voltage ramping speed was approximately 0.35 V/s. However, they were not precisely controlled in this study. This is because the calibration of the CFBG depends on the synchronized IR data that continuously monitors the temperature profile along the fiber. The position and temperature of the hot wire were not used for calibration. Since the fiber optics has a small size, we assume that its temperature reaches a steady state faster than the 80 μ s exposure time. A description has been added and highlighted in yellow in the manuscript.

Changes made in the third paragraph of Sec. 2.3 (page 12):

The hot wire's moving speed, both in the horizontal and vertical directions, and the voltage ramping speed were approximately 0.25 mm/s and 0.35 V/s, respectively. However, they were not precisely controlled in this study. This is because the calibration of the C-FBG depends on the synchronized IR data that continuously monitors the temperature profile along the fiber. The position and temperature of the hot wire were not used for calibration. Since the fiber optics has a small size, we assume that its temperature reaches a steady state faster than the 80 μ s exposure time.

(Q4) In Figure 8(a), 8(b), since the authors acknowledged that “C-FBG only occupies a portion smaller than IR camera field of view”, then why is there any data at all outside of the CFBG, where no grating exists thus no reflection signal? If the result is consistent, shouldn't that area after CFBG cutoff be blue instead?

Authors' Response:

We truly appreciate the reviewer's concerns regarding the locations that C-FBG does not cover. In this work, the machine learning demodulation was based on vector (spectrum intensity) to vector (thermal profile) correlation, not point (specific wavelength) to point (specific location) correlation. Thermal profile, in other words, temperature distribution, is a continuous curve. Therefore, the spectrum data generated from the C-FBG also contains the thermal information adjacent to it. A clarification has been added to the manuscript.

Changes made in the second paragraph of Sec. 2.4 (page 15):

Examples of thermal profiles for selected time steps are shown in Figures 8c to 8d, where heating location change can be clearly observed. As these three figures show, the demodulated thermal profile nicely matched the experimental IR-collected data, especially in the center region (between two dashes). This is because the C-FBG only occupies a portion of the IR camera field of view, making thermal demodulation of the outer region less accurate. There are data outside of the C-FBG covered range because the machine learning demodulation was based on vector (spectrum intensity) to vector (thermal profile) correlation, not point (specific wavelength) to point (specific location) correlation. Thermal profile, in other words, temperature distribution, is a continuous curve. Therefore, the spectrum data generated from the C-FBG also contains the thermal information adjacent to it.

(Q5) I find it confusing that the 28.8 μ m/pixel spatial resolution is directly coming from the IR camera. Due to the neural network calibration process mentioned in the paper, how did the IR camera spatial resolution translate into the sensors spatial resolution? Why is it 1:1?

Authors' Response:

We genuinely appreciate the reviewer's concerns regarding determining the spatial resolution. As explained in Q4, this work trained a machine-learning model to demodulate the C-FBG signal by correlating the spectrums with thermal profiles. Such a model takes the spectrometer spectrum data as the input and the thermal profile observed by the IR camera as the output. Therefore, when using this model, the output will always retain the same format and the same physical meaning, which is the thermal profile of that given length, and the resolution is retained as the IR camera's. It is worth mentioning that the spectrometer has a much higher spatial resolution than the IR camera. Therefore, it will not become the bottleneck. Using a higher-resolution spectrometer may potentially improve the model accuracy, but it cannot improve the

output spatial resolution since the IR camera's resolution constrains it. A description has been added and highlighted in yellow in the manuscript.

Changes made in the first paragraph of Sec. 2 (page 7):

The spatial resolution is only limited by the IR camera's spatial resolution. This is because when using the trained model for demodulation, the output will always retain the same format and share the same physical meaning and resolution as the thermal profile collected by the IR camera. Note that the spectrometer has a higher spatial resolution than the IR camera. Therefore, it will not become the bottleneck.

REVIEWERS' COMMENTS

Reviewer #2 (Remarks to the Author):

All my comments were addressed, nice job.